# Discovery and Classification of the φ6 Bacteriophage: An Historical Review

**DOI:** 10.3390/v15061308

**Published:** 2023-05-31

**Authors:** Paul Gottlieb, Aleksandra Alimova

**Affiliations:** Department of Molecular, Cellular and Biomedical Sciences, The City University of New York School of Medicine, New York, NY 10031, USA

**Keywords:** bacteriophage, φ6, early discovery, cystovirus, dsRNA

## Abstract

The year 2023 marks the fiftieth anniversary of the discovery of the bacteriophage φ6. The review provides a look back on the initial discovery and classification of the lipid-containing and segmented double-stranded RNA (dsRNA) genome-containing bacteriophage—the first identified cystovirus. The historical discussion describes, for the most part, the first 10 years of the research employing contemporary mutation techniques, biochemical, and structural analysis to describe the basic outline of the virus replication mechanisms and structure. The physical nature of φ6 was initially controversial as it was the first bacteriophage found that contained segmented dsRNA, resulting in a series of early publications that defined the unusual genomic quality. The technology and methods utilized in the initial research (crude by current standards) meant that the first studies were quite time-consuming, hence the lengthy period covered by this review. Yet when the data were accepted, the relationship to the reoviruses was apparent, launching great interest in cystoviruses, research that continues to this day.

## 1. Introduction

The year 2023 marks the 50-year anniversary of the discovery, isolation, and initial classification of the φ6 bacteriophage in the laboratory of Anne Vidaver, PhD, at the University of Nebraska [1]. As the first member of the *Cystoviridae* virus family, it maintained its unique position until 1999 when the laboratory of Leonard Mindich, PhD, at the Public Health Research Institute isolated eight additional species: φ6 to φ13 [2]. Officially seven species are accepted in the *Cystovirus* genus by the ICTV listing: φ6, φ8, φ12, φ13, φ2954, φNN, and φYY (https://ictv.global/taxonomy, accessed on 14 March 2022), which creates confusion as there are additional examples, including recently isolated φZ98, a lytic phage that can specifically lyse liposaccharides (LPS) defective strains of the genus *Pseudomonas* [3]. With the segmented nature of its double-stranded RNA (dsRNA) genome and multi-layered structure, the virion strongly resembled reoviruses, providing a simplified, easily manipulated model for the assembly and replication of the more complex animal virus group [4]. Therefore, during these 50 years, the φ6 bacteriophage has been studied extensively, serving as something of an avatar for many segmented dsRNA viruses in regard to genome packaging, replication, transcription, identification of the component proteins, and culminating with the first analysis of the capsid assembly mechanism. φ6, in possession of an outer lipid envelope, made a significant contribution in studies of viral membrane acquisition and its assembly with proteins. This review covers the major milestones of the research including the isolation and identification of φ6 as a lipid-containing, segmented dsRNA bacteriophage, and the initial descriptions of the molecular mechanisms governing the replication process, a period of approximately 10 years. This period prior to the advent of modern recombinant nucleic acid methodologies and employing early imaging technology nevertheless defined the nature of the virus structure and its replication. The first attempts at ultrastructure analysis will be described as well, which sets the stage for future atomic level reconstruction imaging, research that continues to the present. 

## 2. Discovery

The official discovery and isolation of φ6 was in 1973 by the laboratory of Anne K. Vidaver, PhD, of the University of Nebraska, Plant Pathology Department [1]. The initial sentence of the characterization report publication notes that the “bacteriophage φ6 was isolated during an investigation of bacteriophages of phytopathogenic pseudomonads”; this investigation appears to have taken place 4 years earlier than reported in the 1973 publication, demonstrating an interesting attempt at the literature forensics. According to the principal investigator Anne Vidaver, in her own career review, an initial paper was denied publication by reviewers because no one had ever found a dsRNA bacterial virus before, and the result was met by disbelief. As Dr. Vidaver related, “the paper is now part of scientific obliteration and is not cited” [5]. In 1973 the definitive paper was published, reporting newly discovered φ6 as a lipid-containing dsRNA virus with *Pseudomonas syringae* pv. *phaseolicola* as the host bacteria [1]. The virus was isolated from *P. phaseolicola* infested bean straw using *Pseudomonas syringae* pv. *phaseolicola* HB10Y to prepare enriched culture. 

The basic biochemical and biological properties were determined using the molecular tools available one-half century past. Density gradient analysis in both sucrose and CsCl demonstrated that distinct zones of absorbance at 260 nm coincided with infectivity, measured as plaque forming units. The CsCl buoyant density was calculated at 1.27 g/mL, which was similar to the well-studied lipid-containing phage PM2. The stability of the viral particles was tested under one-step growth conditions using either nutrient broth (NBY) or semisynthetic medium (SSM). The latent period was shorter in SSM (80–155 min in SSM vs. 120–160 min in NYB). The burst size in SSM was almost twice that in NYB, 250–400 vs. 125–150, respectively. The purified bacteriophage was most stable at 0 °C. The Nebraska group was able to isolate *P. phaseolicola* host cells resistant to φ6, and demonstrated with fluctuation analysis and re-spreading tests that the resistant bacteria resulted from random mutations and were not induced by exposure to φ6 [6]. However, what was of the most interest was the extreme sensitivity of the bacteriophage to organic solvents and sodium deoxycholate, suggesting that the particle contained a lipid. The φ6 lost infectivity in the presence of phospholipase A. The fatty acid analysis of chloroform-methanol-extracted and methylated fractions indicated a similarity to that of the host bacterium, *Pseudomonas syringae* pv. *phaseolicola* HB10Y. Overall, the bacteriophage was determined to be composed of 25% lipid, 62% protein, and approximately 13% RNA. 

The first electron microscopy (EM) of negatively stained bacteriophage particles using a mixture of potassium phosphotungstate and vanadatomolybdate clearly demonstrated an outer membrane surrounding a polyhedral core estimated to be approximately 60 to 70 nm diameter (Figure 1A). Additionally, the EM imaging showed that φ6 attached to the host cell pili by what was reported to be a membranous structure (Figure 1B). In what must have been a staining or glutaraldehyde fixation artifact a “saclike” appendage was visible, its length and width are variable in several images. Electron micrographs published by Ellis and Schlegel in December 1974 (Lilly Research Laboratories, Indianapolis, IN, USA) of phosphotungstic acid (PTA) stained φ6 (samples supplied from the Vidaver lab) showed structures with a dense inner core and some particles had a blunt tail-like structure, interpreted as “streaming” lipid layer [7]. Chloroform-treated particles that had lost the envelope and P8 matrix appeared as hexagon shapes in the projection images and the authors speculated these to be icosahedral shapes; in actuality, they were later revealed to be dodecahedrons (Figure 2A). The similar hexagon shaped structures were observed for Triton-X100 treated bacteriophages. Cell sections of infected HB10Y indicated that the virosome was central in the host cell and not near the cell wall (Figure 2B,C). 

## 3. φ6 Was Found to Contain Three dsRNA Segments

The φ6 initial classification report clearly identified the bacteriophage as containing RNA but made no mention of the genome’s double-stranded and segmented qualities. A full description of the segmented dsRNA is found in a second study by the Vidaver laboratory, also published in 1973 [8]. The results were derived utilizing the conventional techniques of the era and were quite convincing in describing the unique RNA qualities of the φ6 bacteriophage. The base pairing demonstrated A/U and G/C ratios of approximately 1 that suggested the dsRNA conformation. A rapid rise in hyperchromicity was characteristic of dsRNA vs. single-stranded RNA (ssRNA) that contained only a secondary structure. The dsRNA nature was double checked by an equilibrium density gradient in Cs_2_SO_4_ showing monodisperse banding estimated at 1.605 gm/cm^3^. Furthermore, the elution of the RNA from methylated albumin kieselguhr (the MAK column) demonstrated a homogenous species at 0.78 M NaCl. 

Rate zonal sedimentation using linear and log sucrose density gradients separated three distinct dsRNA components estimated to be 17.0, 15.5, and 14.5 S. The double-stranded nature of the RNA was confirmed by showing resistance to pancreatic RNAase, spleen phosphodiesterase, and partial digestion by snake venom phosphodiesterase, qualities similar to those observed with reovirus dsRNA [9]. Further proof of the segmented nature of the φ6 dsRNA was achieved by the separation using electrophoresis in polyacrylamide/agarose gels. Finally, the EM was employed on formamide-treated preparations of the RNA using the Kleinschmidt method [10,11] and three distinct dsRNA molecular lengths were visualized in ephemeral shadow images, from which the sizes could be calculated. Of peripheral interest to the classification of φ6 as a dsRNA containing entity was the consideration of its utility as a source dsRNA for the induction of interferon. At the time, interferon was being developed as a broad spectrum antiviral pharmaceutical and the simple isolation of large quantities of dsRNA was considered to have considerable potential as a reagent in the industrial process [12]. The relative abilities of each of the three segments was checked for the inducement of interferon and the amount of units induced increased from the small to large segment [13]. The research was published in July 1974 and 1 month earlier patent number 3,819,482 “Method of preparing high yields of double-stranded ribonucleic acid” was issued by the United States Patent Office. Of interest is that the patent claims described only the reagents and the isolation method of the dsRNA as φ6 being a natural microbial agent is not subject to intellectual property.

## 4. First Identification and Initial Characterization of the φ6 RNA Polymerase

The initial classification of the φ6 bacteriophage continued at the Plant Pathology laboratory with the demonstration that an unknown enzyme tentatively assigned as RNA replicase catalyzed the incorporation of ribonucleoside triphosphates into the dsRNA genome [14]. In the absence of the specific identification of the enzyme, the polymerization assay relied on the crude extraction of enzymatic activity from gradient isolated bacteriophage particles. Later, the enzyme was identified as an RNA-dependent RNA polymerase (RdRp) P2 protein [15]. If the purified bacteriophage was used in the polymerization assay, the low enzymatic activity was observed. The following bacteriophage treatments were conducted to achieve consistent and reliable results. Since the lipid envelope was a likely barrier to substrates, the bacteriophage particles were treated with organic solvents and detergents to disrupt the envelope and possibly stimulate enzymatic activity. The use of organic solvents and detergents produced inconsistent results in regard to the polymerase activity. RNA polymerase activity in reoviruses can be stimulated by a short heat treatment and the same idea was applied to φ6, even without any prior knowledge of any similarity between reovirus and φ6. The approach worked and the stimulation of RNA polymerase was observed for φ6 particles after the short heat shock followed by chilling. A particle disruption was observed and confirmed by a broader sedimentation peak after the heat treatment as the partial particle disruption exposed the RdRp protein P2 facilitating the enzyme activity [16].

In many aspects, the RdRp of φ6 was similar to RNA replicase from reoviruses: the overall kinetics of the reaction; the incorporation of ribonucleotide monophosphates into dsRNA; the enhanced effectiveness of Mn^2+^ compared to Mg^2+^; and the extra ss- or ds- exogenous RNA has no effect on the polymerase reaction. All the conclusions were derived based on indirect observations: isotope labeled ^3^H-UMP incorporation. The endogenous double-stranded genome served as the RNA template [17,18]. The RNA polymerase reaction was linear for approximately 10 min and then plateaued, but the reaction was stimulated post plateau if an additional heat-treated bacteriophage was added. If RNase was applied at the reaction start, the ^3^H-UMP incorporation rate significantly decreased but if the enzyme was applied at the reaction conclusion, no diminishment of isotope incorporation was noted. Since ribonucleases cannot enter the assembled nucleocapsid, the nascent transcripts were likely immediately digested. Pyrophosphate but not orthophosphate substantially inhibited the reaction.

An indirect demonstration of the association of the RdRp with the particle was shown using density gradient analysis of heat shock activated purified bacteriophage. The absorbance at 260 nm (A260) for the most part coincided with enzyme activity indicating the protein was integral to the bacteriophage particle. (Later in 1976, the Nebraska group demonstrated the association of the RdRp with the NC particles, which were isolated from the entire virus particle using nonionic detergents [19].) The 1973 studies continued with an analysis of the nature of the viral RNA synthesized by the differentiation of the dsRNA from ssRNA synthesis. The fundamental issue was whether the incorporation of isotope labeled UMP was into ssRNA, dsRNA, or both. The RNase hydrolysis of the polymerized RNA was used to determine the degree of ^3^H-UMP incorporated into the dsRNA. The quantitative assays indicated that the incorporation was greater in the M and S segments and the CMP incorporation was considerable higher than the AMP incorporation. A hierarchy of incorporation of each ^14^C isotope-labeled NMP was noted: CMP being highest followed by GMP, UMP, and AMP—at this time providing the only indirect indication that the three RNA segments had different base sequences, at least in the 5′ region.

## 5. RNA Conformation

The Plant Pathology research group next considered the physical nature of the dsRNA segments in regard to the base compositions as determined by hybridization studies [20]. The major question at this stage of the studies was whether the two smaller dsRNA segments of the φ6 genome were merely derived from the large dsRNA segment (perhaps at a specific cleavage site). Based upon the estimated sizes of the segments, this was certainly a plausible assumption as the size sum of the S and M segments approached that of L. However, once the dsRNA segments were denatured and quickly cooled, there were only six bands that migrated in PAGE. Had any cross-annealing occurred among the l, m, and s ssRNA segments, more than six A_260_ peaks would have been obtained. In a second demonstration that the three segments were unique, ^32^P-labeled dsRNA segments were annealed with ten-fold excess of unlabeled unfractionated ssRNA and each strand annealed only with strands of the same size. The indirect analytic methods used a half century ago lacked proper resolution, which led to erroneous conclusions: the dsRNA segments likely had limited single-stranded tails. These methods were based on ribonuclease digestion combined with analytic gradients. Iba et al., 1982, determined the 5′ and 3′ terminal sequences of the plus and minus strands of the dsRNA segments, showing that they were flush, lacking any ssRNA tails [21]. The methodology made use of partial RNase digestion (T1, U2, PhyM, or *B. cereus* RNase) and partial hydrolysis. Two-dimensional gel electrophoresis of end-radiolabeled and denatured RNA segments allowed the identification of the terminal sequences of the three genome segments, showing that M and S began with GG while L started with GU. An insightful prediction noted that the common sequence was likely to contain the recognition site for the RdRp, an observation later to be confirmed by sequence analysis of the entire genome. 

The major question at this early stage of φ6 research was whether the ssRNA was synthesized in a conservative or semiconservative mechanism and if it was functional as a translational message. In papers published from 1975 to 1976, the Plant Pathology group first identified and described the specific properties of the classes of viral RNA and the intermediate forms [22,23]. The ssRNA were identified as the precursors to the dsRNA genome, transient in nature, and also functioning as messenger RNAs (mRNAs). The term plus strand was used for the mRNA segments, and in this regard the mechanism of RNA replication was seen to be similar to the reoviruses. However, in contrast to the reoviruses, it was evident that the φ6 mRNA segments had to be polycistronic to accommodate the estimated 8 to 10 structural proteins of the bacteriophage. The study also noted that the plus strands are not synthesized in constant proportions in that the late phase favored the production of the s and m segments. Precisely which proteins were required early in the replicative cycle could only be speculated at the time and the answer awaited genetic studies. Yet an insightful prediction was based on the observation that the replicative intermediate was rapidly labeled in both ssRNA and dsRNA, implying that the φ6 transcription was likely semiconservative, differing from the reovirus conservative mechanism [24]. The dsRNA synthesis was found to be independent of host cell function, as rifampin and chloramphenicol addition to the infected HB10Y cultures did not inhibit RNA synthesis. In furtherance of the proof of semiconservative RNA synthesis, Rimon and Haselkorn in 1978 found in long-term pulse-chase assays that the isotope label flowed from dsRNA intermediates to completed ssRNA and then back to dsRNA [25]. The mechanism had to be semiconservative because each progeny dsRNA contains one parental strand and one new strand (Figure 3). The figure schematically indicates that transcription in φ6 is via a strand displacement reaction where each new plus sense “pushes off” the previous strand. Therefore, the first displaced transcript would be unlabeled while the subsequent strands would incorporate the label. Next Partridge et al., 1979, utilized isolated NCs as the enzyme source in an in vitro transcription assay and determined the RNAase sensitivity of the products in order to distinguish the ssRNA and dsRNA [26]. The polymerization reaction appeared to proceed through a replicative intermediate-RNA (RI-RNA) that consisted of a full-length strand and a partially completed ssRNA transcript. The earliest RNA synthesized was RNAase resistant, indicating that it was likely dsRNA. Van Etten, 1980, provided direct evidence that φ6 utilized semiconservative transcription and that the large ssRNA was synthesized at a lower amount [27]. Prelabeled NC dsRNA was produced by growing the bacteriophage in ^3^H-uracil containing medium. When the NC RNA polymerase assay was initiated in vitro, the radioactive label was displaced into the three ssRNA plus segments, clearly indicating that the mechanism was semiconservative. Next Coplin et al. found that the plus sense ssRNA were intermediates for the replication of the genomic dsRNA segments [22]. Finally, Cuppels et al. of the Nebraska group showed that in vitro translation of the ssRNA produced φ6 proteins, i.e., large segment codes P1, P2, P4, and P7, medium-size one codes P3, P6, and P10, and the small segment codes P5, P8, and P9 [28]. 

The first study on bacteriophage φ6 by the laboratory of Leonard Mindich at the Public Health Research Institute of New York (PHRI) identified most of the virus proteins using ^14^C-leucine labeled bacteriophages [29]. The bacteriophage was of interest both in regard to describing interactions between lipids and proteins during membrane biosynthesis and of the utility in understanding the mechanisms of dsRNA virus assembly. The protein components from purified bacteriophage particles were identified by electrophoresis in SDS-PAGE [30]. ^14^C-leucine was the label and the amino acid was assumed to be uniform weight fraction in each protein; therefore, this early estimation was admittedly imprecise. Nevertheless, the SDS-PAGE autoradiograms provided a reasonable identification of the proteins, which were numbered according to the electrophoretic migration rate. Sinclair et al. demonstrated the proteins isolated from the bacteriophage as compared to an infected cell lysate control preparation (Figure 4). The clarity of the analysis was dependent on the addition of rifampin. The rifampin added to the infected cells reduced the host protein background by diminishing the bacterial but not viral RNA polymerase function, simplifying the bacteriophage protein identification. The autoradiogram clearly shows ten easily identified proteins that compose the bacteriophage particle. The proteins were labeled from P1 to P10 according the migration order. The infected cells showed two additional nonstructural proteins designated P11 and P12. (Notably when new cystoviruses types were discovered in the future, the protein nomenclature was maintained in the interest of uniformity even when the SDS-PAGE migration rates differed from φ6 [2].) The origin of P11 is still controversial as it is considered a possible precursor of P5. The SDS-PAGE autoradiogram of the infected cell lysates showed that P11 migrated at a slightly higher position than P5. When the label was switched to ^35^S-methionine proteins P5 (11), P9, and P10 were not labeled, indicating they lacked this amino acid. The 1975 study also made initial observations in regard to the temporal regulation of the bacteriophage protein synthesis. The host cells were UV irradiated prior to infection, markedly reducing overall protein synthesis yet proteins P1, P2, P4, and P7 production remained at a significant level. Pulse labeling assays showed that these four proteins appeared approximately 10 min after infection, but the other proteins did not appear until 45 min. Triton X-100 treatment extracted proteins P3, P6, P9, and P10, indicating that the four proteins are associated with the bacteriophage envelope. After the Triton-X 100 removal of the envelope protein, P8 was next extracted with guanidine hydrochloride, indicating a second specific layer below the envelope. (Of interest was the misinterpretation of the P8 maturation based on the pulse labeling assays. The protein seemed to appear as a double in the cell lysate autoradiograms and single in the proteins isolated from the bacteriophage particles. Later structural studies would show that P8 had different conformations when assembled into the matrix layer, consisting two states of interdigitating P8 trimers that are designated as either open or closed, possibly accounting for the initial erroneous interpretation [31]). The physical effect of the addition of different concentrations of Triton X-100 were imaged by electron microscopy in phosphotungstic acid (PTA) negatively stained samples, showing progressive removal of the envelope and exposed nucleocapsid. The isolated procapsids (PC) appeared as hexagons in the projection view, indicating that approximately 1% of Triton-X100 disassembled the P8 layer. 

At the time the Mindich group published their initial paper, the Nebraska group was writing a paper that described similar results in the identification of the protein components of the entire bacteriophage and the NC [19]. The nonionic detergent isolation of the NC from its surrounding envelope and SDS-PAGE analysis of the migration rates of the proteins allowed an estimation of the molecular weights. While this group also counted the ten major proteins, they acknowledged that the molecular weight estimates tended to be approximately 20% higher than that of Sinclair et al. [28], explained as differences in the sample treatment. The precise values would have to await future genome sequencing studies. The study included the iodination of the entire bacteriophage and the NC using ^125^I, showing that P3, P5, P6, P9, and P10 were on the outer surface and P8 was located on the surface of the nucleocapsid. At this time, the basic biochemical investigations were complete. 

## 6. Isolation of φ6 Mutants: Specific Identification of Protein Function

Four papers from the Mindich laboratory were all accepted for publication in July 1976, published back to back in order to define the φ6 gene products and describe the temporal regulation of translation [32,33,34,35]. It needs to be noted that the sequencing of the φ6 RNA was started in 1986 [36] and not completed for several years, so the following work was conducted without knowledge of the actual gene sequences. The strategy required the creation of amber mutations in bacteriophages and suppressor strains of the cells for the mutants. The suppressor strain should support the growth of bacteriophage mutants as well. This ornate and very labor-intense strategy allowed the analysis of the translation decoding mechanism of the phages. The plasmid pLM2, which bears an amber mutation in ampicillin (amp) and tetracycline (tet) resistance genes, was created on the basis of the PR1 plasmid that carried the genes for kanamycin (kan), amp, and tet resistance and is capable of replicating in multiple gram-negative bacteria.

Unfortunately, the isolation of the suppressor mutants of the natural host of φ6, *Pseudomonas syringae* pv. *Phaseolicola* HB10Y (HB) was not successful [37], compelling efforts to find an alternative host bacterium that supported φ6 replication with suppression capability. The source of the new host was water taken from the East River in New York City. Out of several hundred colonies grown from river water, only two strains were found to support the replication of φ6. It was described as a rare and fortuitous result by the study authors. One strain chosen for additional study was designated ERA and had characteristics identical to *P. pseudoalcaligenes*. Indeed, initially φ6 replicated on ERA at a very low efficiency of plating (EOP) so the progeny bacteriophage was selected, named as φ6h1, and had an EOP on ERA of 0.6. Interestingly, the EOP of φ6h1 in HB was comparable to the EOP in ERA. Plasmid pLM2 was transformed into ERA and a stable transformant was selected. The selected strain could support the growth of bacteriophage PRD1 that contained an amber mutation. This next generation of *P. pseudoalcaligenes* ERA (designated ERA (pLM2) S4) also supported the replication of φ6h1, although at very low EOP. 

Using NTG and 5-fluorouracil (FLU) mutagenesis of φ6h1s nonsense mutants were selected and mutants with a low reversion frequency were retained for study. After infection of the nonpermissive HB host, bacterial cell lysates were screened for viral protein synthesis compared to a wild-type control infection. In total, 62 of the mutants were assigned to 8 nonsense mutant classes, based on missing proteins in autoradiograms of SDS-PAGE (Figure 5A) and the analyzed data presented in Figure 5B. Of the ten structural proteins, P4, P7, and P10 were not represented by any nonsense mutants. There were some suggestions why these three proteins were not represented but none were conclusive. The temporal regulation of protein synthesis was surmised as P1 and P2 mutants were unable to synthesize late proteins and were clearly associated with bacteriophage dsRNA synthesis. Specifically, the mutant classes that deleted P1 and P2 were incapable of synthesizing any segment of the dsRNA genome, indicating the requirement of these two proteins to constitute a polymerase complex [35]. A temperature-sensitive (*ts*) mutant was also isolated (designated *ts10*) that mapped into the same genome segment as the 1 and 2 class, indicating it was likely polymerase associated and at the nonpermissive temperature no dsRNA could be synthesized. The authors speculated that the dsRNA genome was likely replicated through an ssRNA intermediate as the transcripts could not be synthesized early in infection in the class 1 and 2 mutants.

Mapping of the mutations allowed a further delineation of the genetic organization of the φ6 genome. In total, 46 *ts* mutants were selected and, along with the set of nonsense mutants, organized into three linkage groups, designated A, B, and C, with the goal of identifying the genomic location on one of the three dsRNA segments [33]. Set A mutations were seen to eliminate P1 and P2 and consequently all the late proteins. Since the P1 and P2 molecular weights had been estimated by PAGE, the minimal genome that carried the genes size could also be approximated at 3.4 × 10^6^ D for the size of the dsRNA. The value was in the ballpark for the established dsRNA segment size determined by the Plant Pathology researchers in Nebraska [8], suggesting that only the largest component of the genome could contain the genetic information encoding proteins P1 and P2. Mindich et al. [33] went on to speculate that the genes encoding P4 and P7 were also located on the large dsRNA segment, in the realization that the nucleocapsid core proteins were genetically clustered, an observation confirmed years later by cDNA sequence analysis [38]. Linkage set B was seen to contain the genes encoding P3 and P6, which when mutated prohibited host cell adsorption. At roughly the time of the genetic studies, Wanda et al. published a study that showed butylated hydroxyltoluene (BHT) inactivated φ6 by inhibiting the particle attachment while leaving the bacteriophage envelope intact, providing a biochemical reagent that could be exploited to understand the attachment mechanism [39]. Indeed, BHT was later shown by Bamford et al. to selectively remove P3 from the bacteriophage surface [40]. Finally, in regard to linkage set B, the P10 gene position was not identified in the absence of a mutant and would need to await the sequence information. In fact, the genetics paper mistakenly predicted it would be located in set C. Set C carried the genes for the membrane associated proteins, P9, P5, and P11. The nonstructural protein, P12, that is responsible for the bacteriophage membrane assembly was included in the linkage group as was the NC surface lattice protein P8. Therefore, this genome clustered the genes that formed the two outer layers of the particle. Within the discussion, the authors noted that there was no evidence for intramolecular recombination, but speculated that it could take place at a frequency too low to detect—foresight as future work demonstrated intramolecular recombination among the three genome segments [41].

Of great interest was the observation that polar relationships existed among some of the late proteins. Prior to the sequence analysis, these polar relationships were hypothesized to be a consequence of mRNA secondary structure. P8 mutants are missing P12 while the loss of P12 was not seen to diminish P8. The reason was later resolved when genome sequence information was obtained demonstrating that the P12 gene lacks a Shine–Delgarno (SD) sequence making it translation coupled to P8 [42]. A similar polar relationship was noted for proteins P9 and what was then termed P5/11 in that mutants missing P5/11 synthesized normal levels of P9, but when P9 was lost so was P5/11. Again, later sequence analysis resolved the phenomenon by showing translational coupling of P5/11 to P9 [36]. However, the polar relationship between P6 and P3 presented a greater conundrum that is still to be entirely resolved. The genes are located on the middle (M) dsRNA segment and nonsense mutants in gene 3 that produced a P3 fragment show a small diminishment of P6. Oddly, class 6 nonsense mutants considerably reduced the P3 level, as visible on the autoradiograms. P6 and P3 interact with each other in the bacteriophage envelope and the thought at the time of the mutant studies that it was possible that each affects the stability of the other. Additionally, one nonsense mutant of P6 slowed the SDS-PAGE migration rates of P5/11, but oddly when a P6 reversion was isolated, which was expressed in HB, P5/11 still migrated slower. In another aberration, a reversion of mutant class 9, 5/11 caused P5 and P11 to migrate at a faster rate in the SDS-PAGE. The mutants analyzed in this first of the four paper series [34], setting the stage for greater detail of the viral replication and morphogenic mechanisms. Continued searches for additional nonsense mutants resulted in a P7 mutant that was seen to be polar on P2 located in the large L segment linkage group [43]. As with the other φ6 polar pairings, the distal gene made a significantly reduced amount of protein compared to the proximal. Later sequence analysis demonstrated that the P2 gene lacked an independent SD sequence and was dependent on a translational readthrough from P7 [38]. Rimon and Haselkorn at the University of Chicago, Department of Biophysics and Theoretical Biology utilized conditional lethal *ts* mutants to analyze the genetic arrangement of the RNA replication and transcription apparatus in 1978 [25]. The methodology employed the selection of NTG induced mutants of φ6 that were nonpermissive for growth at 28 °C in HB10Y and assigned to three complementation groups, the results correlating with those of Mindich and Sinclair [21]. The study focused on two mutants designated, *ts51* and *ts81*, that inactivated the RNA synthesis mechanism at the nonpermissive temperature. As with the results reported by Mindich and Sinclair, the blockage of RNA synthesis by the mutants prevented the synthesis of late proteins as determined by isotope labeling and SDS-PAGE.

## 7. The First Models of φ6 Morphogenesis and Architecture

The Mindich laboratory utilized additional NTG selected nonsense mutants of φ6h1s in order to model the morphogenesis of the bacteriophage particle [32]. This study only employed late proteins in the analysis. The genes encoding the early proteins P1 and P2 were not included in the study. The double-labeled (^3^H-glycerol and ^14^C-leucine) bacteriophage-like particles were isolated from the gradients and the missing proteins were noted by SDS-PAGE. The results were summarized in the schematic model reproduced from the 1976 paper (Figure 6). In this initial morphogenic model, the entire inner core assembly was presumed to initiate from P1 encompassing all except the lipid layer and the integral proteins. Lipid acquisition was seen to be protein P12 dependent and the final viral maturation resulted by the addition of the P6/P3 attachment apparatus. The study discussion acknowledges the aforementioned EM images that suggested a pleomorphic surface structure, yet the possibility that the attachment proteins are distributed over the particle surface was now strongly suspected. Therefore, a detailed morphogenesis description required better imaging by EM of the bacteriophage particles. 

The early attempts of ultrastructure analysis of φ6 were performed by Bamford et al., 1976, noting a 65 to 75 nm particle that appeared to have a tail-like structure of variable length—clearly, this misinterpretation of the images (or staining artifact) remained a source of confusion [44]. Sectioning of pelleted bacteriophage showed a layered structure, i.e., an inner dense core 30 nm in diameter, and another electron dense shell ~50 nm in diameter, which appeared as a dark circle on the micrograph and bi-lipid membrane of ~7.5 nm thickness (Figure 7). When Triton X-100 was applied to the particles, a 50 nm core was exposed. The study confirmed pilus binding as strings of bacteriophage can be seen extending from the host cell although the pili are not evident in the negative stains. The different stages of fusion of the bacteriophage bilipid membrane and the outer membrane of the host cell were captured, but again the tail artifact was seen and described as initiating the infection.

EM studies by Bamford and Mindich working together at PHRI looked at sectioned and stained host cells infected with φ6h1s nonsense mutants. The cells were harvested at several time points after infection and examined for particle morphology, further confirming the assembly model presented in Figure 6 above. Most significantly gene 8, 12 mutants could still package dsRNA (seen as internal particle density), only PC, not NC, was formed. It was in this study that the designation “procapsid” was first used—the NC precursor. Bamford et al., 1982, in a short note continued the description of the infection process using additional EM techniques [45]. Scanning EM images confirmed that the bacteriophages attach to the host pili. Freeze-fracture EM analysis demonstrated a bulge once the bacteriophage envelope fuses with the host. The Mindich laboratory was able to isolate a 120 S particle from lysed host cells and found that they consisted of proteins P1, P2, P4, and P7 [46]. EM images showed hexagon shaped, ten-point quasi-circular, and “star-like” conformations—now recognized as empty PCs, each rotated in one of three orientations. An estimate of the number of molecules of each component in the virion suggested approximately 100 P1, 14 P2, 100 P4, and 80 P7 [47]. It would later require more contemporary studies to demonstrate accurate values: 60 dimers of P1 formed the framework of the PC and P4 was a hexameric NTPase that could be located at the 12 faces of the dodecahedron if the positions were fully occupied [48]. The precise positions and occupancy of the P2 and P7 proteins of the cystoviruses remain elusive and might depend on the stage of the replication cycle and assembly of the bacteriophage particles [48,49,50]. 

The first suggestion that the φ6 NC is an icosahedral shape was made by the Nebraska Plant Pathology group working with sections of infected host cells [51]. After glutaraldehyde fixation with post-fixing in OsO_4_ and uranyl acetate, they described the dsRNA genome as hexagonal ring termed “doughnut shaped” due to a distinct central open area. The entire particle was measured at 75 nm diameter and the NC at 60 nm; reasonable accuracy for the methodology used. Significantly, the NC was recognized as having an icosahedral architecture. Two papers published collectively by Steely, Lang, and Yang clearly defined the PC as a dodecahedron. PTA stained NCs highlighted the PC structure and the three morphological forms were noted, the hexagon, ten-pointed circle, and stars, consistent with icosahedral symmetry and describing a regular dodecahedron. Goniometer tilt angles further confirmed that the stained projection images were consistent with a dodecahedral framework [52,53]. Again, it must be noted that the PTA staining obscured the P8 matrix and future work would demonstrate that this outer covering was icosahedral. Indeed, recent single particle reconstruction of cryo-electron microscopy images showed that (at least with cystovirus φ12) there are two layers, the P8 matrix over the PC. The P8 layer is best described as an incomplete T = 13 icosahedral lattice and enclosed PC—as T = 1 layer. The symmetry axes of the T = 13 layer superimposed the enclosed T = 1 layer [54].

Lastly, the outer layer of the cystoviruses consists of a lipid bi-layer with embedded bacteriophage proteins and the morphogenesis of the envelope was found to be dependent on the nonstructural protein, P12 [55]. The final part of the assembly process is the envelope acquisition from the host inner membrane—a process that to this day is not entirely understood [56]. The mechanism of envelope acquisition was seen to require the assembly of a lipid component that incorporates the bacteriophage protein P9 (termed the P9 particle) and the P9 particle assembly depends on the presence of the nonstructural protein P12. The Mindich group at PHRI proposed models for membrane acquisition that are shown in Figure 8 [55]. Referral to the figure shows the envelope derived from the host cell inner membrane, which is then associated with assembled NCs. Notably, path II suggests that NC would accumulate near or on the inner cell membrane, but EM studies have never shown this to be occurring in infected cells. In fact, EM imaging usually shows NC more central to the host cell, such as in the carrier state (see the recent review by the authors, [41]) data that seem to favor model number I. Nevertheless, the question remains open and is worthy of future cystovirus research.

In 1979, the Mindich laboratory showed that the lysin is protein P5 and can partially be extracted from the bacteriophage particle with Triton X-100. The group also described a mutant envelope protein P10 that caused a diminishment in host cell lysis and concluded that P5 and P10 need to work together to induce lytic activity [57]. Indeed, Romantschuk and Bamford in Helsinki noted that φ6 resistant mutants of *P. phaseolicola* possessed an altered cell wall structure, rendering them resistant to the lytic reaction [58]. The precise position of P5 in the particle is still not determined; however, later work with cystovirus φ12 suggested a location near the NC type III holes within the P8 matrix [54]. At this position in the matrix, P5 would be partially associated with the bacteriophage envelope, possibly explaining why with detergent only some of the protein was extracted while the remainder stayed within the NC. The Mindich research group then analyzed the assembly of the P3, P6 pili attachment apparatus to the bacteriophage envelope and found that higher growth temperatures affected the amount of P6 placed within the membrane. Since P6 anchors P3, the latter protein was diminished as well.

## 8. Foundational Research Led to a Novel Discovery

The studies described in this review roughly corresponded in time with the establishment of recombinant genetic technology [59]. These new techniques allowed sophisticated research with the bacteriophage to rapidly progress and quickly cDNA cloning with sequencing of the three genome segments was accomplished and assembly studies commenced. At this period the research was dominant in the laboratories of Leonard Mindich at PHRI and of Dennis Bamford at the University of Helsinki. The major contribution from both of the research groups was the detailed description of the RNA packaging mechanism and the structure of the procapsid proteins that mediated the process [4,60]. Of great interest from the final research at the Mindich laboratory and presented as an example of how up-to-date results followed the “classic period” was the discovery that the transcriptional control was governed by host functions, an unexpected observation that occurred by serendipity [61,62,63,64,65] (and personal communication LM to PG, 2008). The work (described in five publications from 2008 to 2013) is worthy of discussion as an illustration of how the foundational research facilitated modern and sophisticated observations. The Mindich research group while studying the φ6 core noted that it contained an additional protein along with the four known proteins, P1, P2, P4, and P7 (remarkedly an overlooked protein in all previous φ6 studies). Figure 9 reproduced from the 2008 publication shows the host cell’s protein YajQ migrating with the PC components that were isolated from carrier state (CS) cells [66,67]. Gene *yajQ* was thought to produce a dispensable protein of unknown function, although it is highly conserved in gram-negative bacteria [68]. 

What proved a most intriguing observation was the role the YajQ played in promoting the L segment transcription. In the φ6 genome, L begins with GU vs. the GG found in the other two RNA segments with the polymerase showing a preference for “G” in the second position. Early in infection L is transcribed in equal amounts with the S and M segments but late in infection S and M are favored over L. Indeed, the research group found that wt bacteriophage plated with low efficiency on *yajQ* knockout host cells but if the L segment contained a start of GG, plating efficiency returned to the normal level. A selected YajQ independent bacteriophage mutant transcribed considerably more l transcripts throughout the replication cycle had mutations in the P1 and P2 gene sequences. Therefore, it was postulated that the role of YajQ was to promote the transcription of L in the early replication phase by binding to the PC, most likely to P1, early in infection. The speculation was that the conformation of the NC is altered by the interaction with YajQ and then the P1–YajQ interaction is transmitted to moderate the P2 polymerase activity by a still undefined mechanism. In furtherance of this concept, the Mindich research group constructed YajQ tagged with green fluorescent protein (GFP) that appeared to show the fluorescence at positions in the infected cells coinciding with the transcribing NCs [64]. The observation was further detailed in a 2013 publication (the final paper from the Mindich research group) that described that the amount of YajQ is limited in the *Pseudomonas* host cells, resulting in only a limited number of NCs able to bind the protein. As bacteriophage replication continues and additional NCs accumulate in the cell cytosol, the L transcription is diminished, accounting for the dominance of M and S transcription. As with the foundational studies, judicious use of selected φ6 P2 mutants enabled transcriptional independence from the YajQ protein [65].

The isolation of φ8 and φ2954 [69,70] also contributed to a greater understanding of cystovirus transcriptional regulation by describing two different and unique mechanisms. The removal, or loosening, of the P8 matrix from the φ2954 NC in order to initiate transcription of the packaged dsRNA was described in 2009 [62]. φ2954 is YajQ independent and chelating agents do not remove the P8 matrix. (The numerical nomenclature for cystoviruses had to jump to higher values as number 29 was approached for reasons that are self-evident [71].) A host protein, glutaredoxinn 3 (GrxC), activates the P2 polymerase and mutations that render this bacteriophage GrxC independent were found to be in the P1 gene. The φ2954 L terminal nucleotide is not G but A (as with φ12 [72]), likely causing the late phase downregulation of its transcription. As in the previous φ6 study, the binding of GrxC to P1 regulates the P2 activity. Bacteriophage φ8 is an outlier in regard to L message regulation in that control of message RNA stability vs. transcription in the mechanism utilized [63]. What is curious about the genetics of this cystovirus is the finding that the first seven nucleotides are identical in the three plus stranded segments, clearly requiring a different mechanism to regulate the relative amounts of the transcripts. In addition, the order of the genes on the L segment is 14, H, 2, 4, 1, and 7 where H is actually a pair of genes Ha and Hb [69]. The unusual gene conformation is shown in Figure 10.

Only protein Hb was actually identified by PAGE while the genes *Ha* and *14* appeared to be dispensable. In spite of the novel gene arrangement at the 5′ end of the L segment and the similarity at the beginning of all three segments, the transcripts of L are still significantly lower late in infection, implying that the regulatory mechanism differs in φ8 vs. other classified cystoviruses. No host protein was found to be involved in the transcription regulation and it was noted that the transcript is diminished due to its degradation. The model described proposed that the Hb protein activates RNaseR to degrade L late in infection—but the possible cooperation with other host proteins could not entirely be ruled out. The temporal control could be explained as the bacteriophage replication cycle only produced the critical level of Hb to activate RNaseR late in infection. The J protein, the gene carried on the small segment, appears to play a role as φ8 mutants with a deletion of its gene produced equal amounts of the transcripts throughout the replication cycle.

The temporal control of cystovirus transcription was the final project to come from the Mindich laboratory. As related in this section it relied on the foundational work that had utilized classical and conventional techniques of genetics, mutagenesis, biochemistry, and microbiology. The fulfillment of the effort relied upon allying these earlier methods and observations with recombinant technology to delineate significant mechanisms extant in cystovirus assembly and replication. Indeed our other review in this Special Issue underscores the progression of the studies in the description of heterologous recombination in the cystoviruses [41]. Yet many questions remain that future research will likely be able to answer using advanced technology that builds upon the previous work. We briefly address the issue below.

## 9. Summary, Conclusions, and Comments on Future Directions

Finally, we note that cystovirus research continues with significant structural analysis of the function and interactions of the virion protein components [30]. We present several examples of ongoing research as illustration—no order of importance is implied. Using electron–cryotomograph Mansha Seth–Pasricha, then working in the laboratory of Jason Kaelber at Rutgers University, New Brunswick, NJ, USA), noted that φ6 can form a single tail-shaped membrane bulge, but only when bound to pili. Purified virions do not exhibit these bulges until they are mixed with cells. The size and shape of the membrane bulge is consistent with the “tail” seen in the original φ6 papers (Mansha Seth–Pasricha, personal communication). The precise activity of lipid transport by the nonstructural protein P12 remains to be determined and is an important question regarding viral envelope acquisition. In particular, the regulation and selectivity of viral RNA packaging was greatly advanced using φ6 as a simplified model for reoviruses. With only three genome segments, the question was more easily approached and amenable to manipulation in a prokaryotic system [73]. The dynamic relationships of the PC protein components were extensively examined, in particular the rearrangement and conformational changes that occurred and accommodated the RNA while packaging [48,49,50,74,75,76,77]. The P4 hexameric NTPase X-ray structure revealed the movements of an RNA binding loop coupled with nucleotide binding and hydrolysis, indicating how ATP hydrolysis drives RNA translocation [78]. There can be no argument that the establishment of φ6 as a significant research model for viral assembly, replication, and dynamic interactions opened a new field in virology and significant mechanisms are still to be explained. It can be anticipated that, with the discovery of additional members of the cystovirus family, better insight on the assembly mechanisms of these viruses will be achievable. Insight into the carrier state as a model for persistent and latent viral infection continues in the authors’ laboratory and those of our collaborators at the City College of New York and the University of Helsinki. We speculate that the recently established study could significantly contribute by providing a model that might extend beyond an academic exercise and prove a useful assay for antiviral agents as well. In regard to evolutionary studies, the cystoviruses, in particular φ6, have been a durable model for fitness studies in regard to host range and the determination of which mutations mediate the process. Highly mutable “hot spots” could be responsible for the expansion in the host range, a process that is known to be of crucial importance in disease causing viruses as well [79].

Finally, it should be noted that the Nebraska group credited Myron Kendall Brakkhe with many contributions to the early φ6 research, yet he refused authorship on any publications as he believed the United States Department of Agriculture, where he held an appointment, would not approve of his participation [80]. This review is best concluded with a quote included in the National Academy of Sciences biography of James L. van Etten “the φ6 virus was worthless as a biological control agent, but scientifically it turned out to be a very unusual virus” [81]. This remark was quite accurate until recently, yet now the bacteriophage could even find utility as a biocontrol in agricultural pathology [82]. 

## Figures and Tables

**Figure 1 viruses-15-01308-f001:**
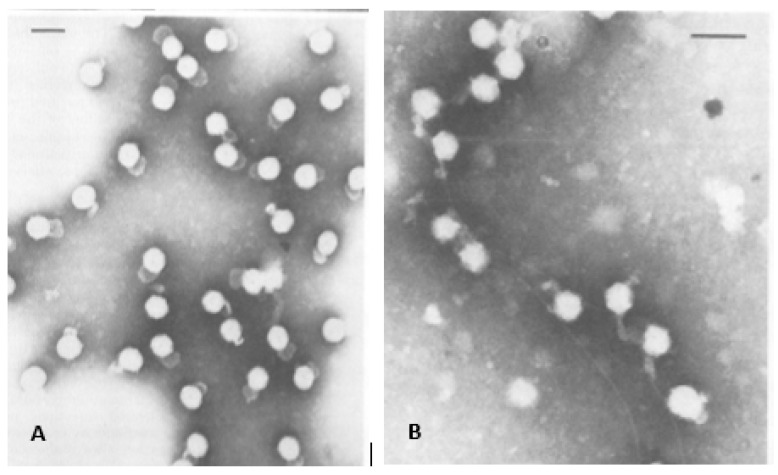
Earliest electron micrographs of bacteriophage φ6. (**A**) The bacteriophages isolated from purified lysate, the envelope structure and “saclike” tail are clearly visible. The “saclike” tail is a preparatory artifact. (**B**) The bacteriophages attached to the pili. The micrographs were used with permission of American Society for Microbiology, from Vidaver et al. 1973 [1]; permission conveyed through Copyright Clearance Center, Inc.

**Figure 2 viruses-15-01308-f002:**
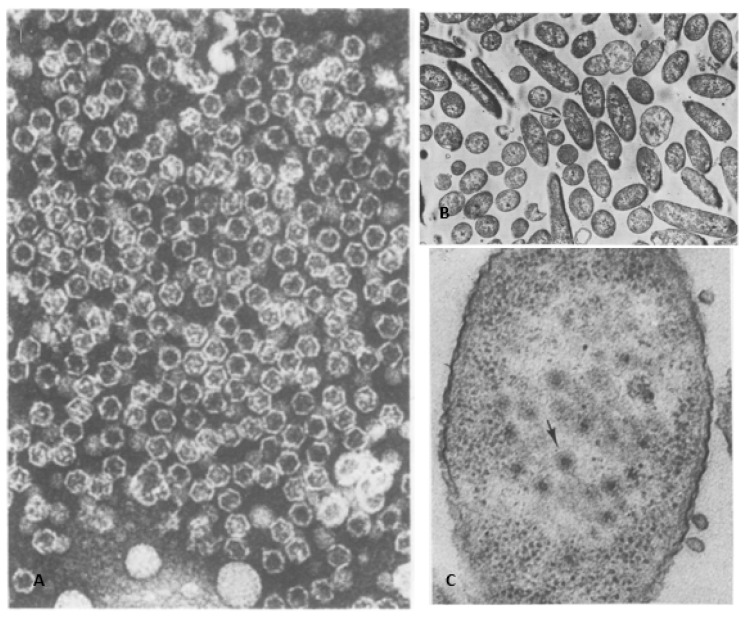
(**A**) The first published electron micrograph of chloroform-treated bacteriophages. The icosahedral nucleocapsids were exposed after outer lipid layer removal. (**B**) A thin section of the φ6 infected *P. syringae* cells prior to the lysis event, 150 min post infection. The samples were stained with UAc. (**C**) Magnified central area of the infected bacteria. The arrow indicates the viruses located in the central part of the cell. Images were used with permission of American Society for Microbiology, from Ellis, L.F. and R.A. Schlegel, 1974 [7]; permission conveyed through Copyright Clearance Center, Inc.

**Figure 3 viruses-15-01308-f003:**
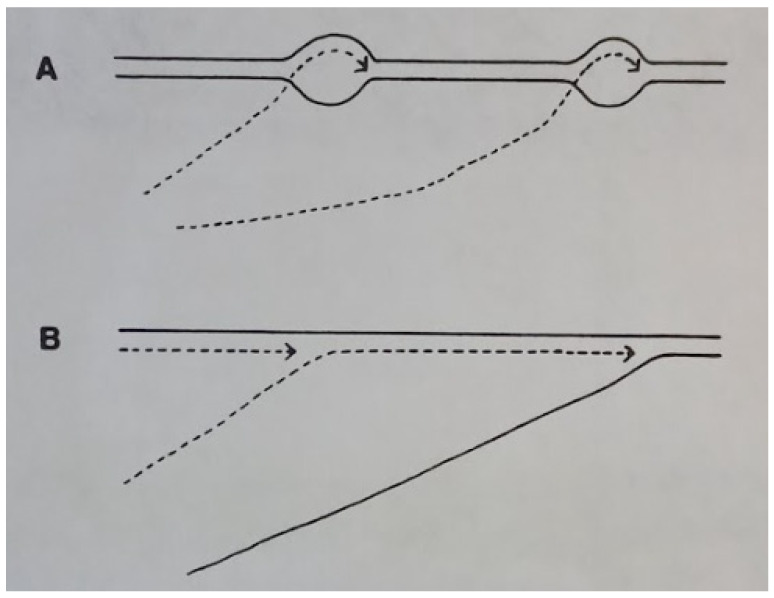
Models for replication of φ6 RNA. (**A**) Conservative transcription, newly synthesized RNA is single stranded. (**B**) Strand displacement, the newly formed RNA replaced one of parental strands. Reprinted from Ref. [25] Rimon and Haselkorn, 1978, copyright BMC.

**Figure 4 viruses-15-01308-f004:**
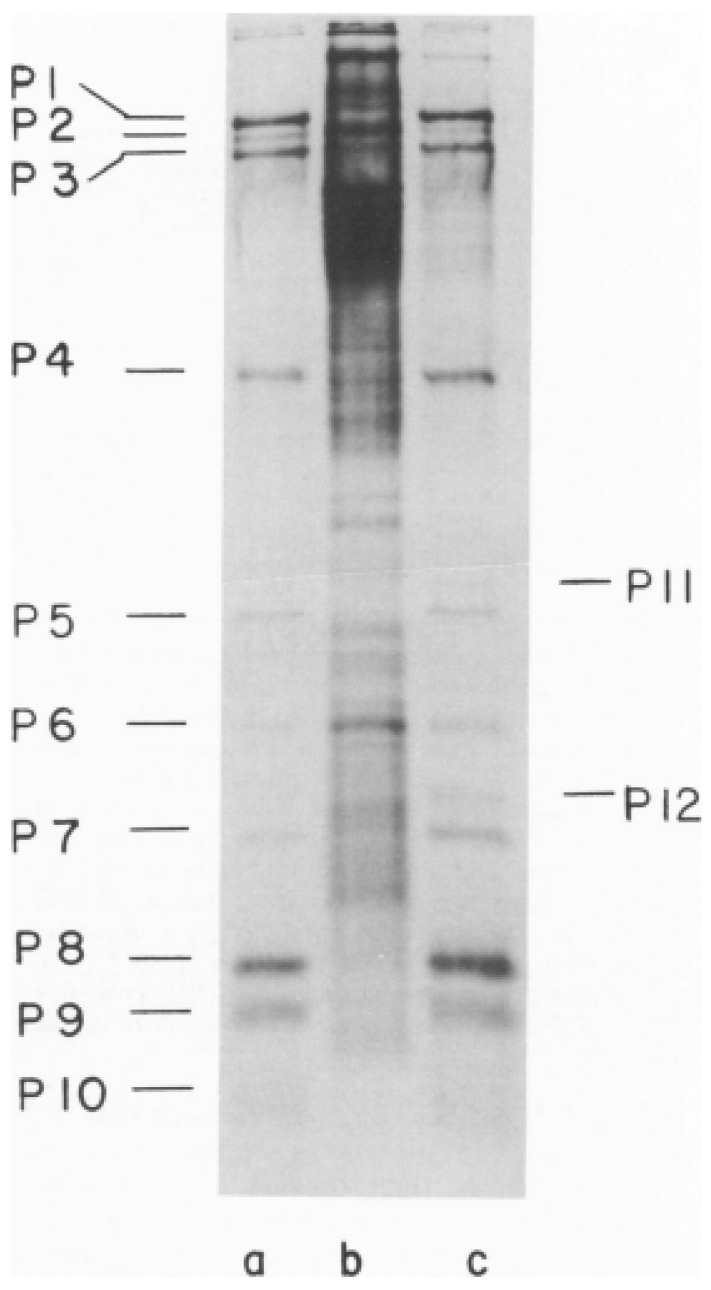
First SDS-PAGE gel showing the φ6 bacteriophage proteins migration rate and assignment of the proteins according to their motility in a 15% discontinuous polyacrylamide gel. The autoradiograph of the ^14^C-leucine labeled samples was exposed for 2 days to allow all the bands to appear. (**a**) Proteins from the purified φ6, (**b**) uninfected cell lysate, and (**c**) rifampin-treated φ6 infected cell lysates. Image is used with permission of American Society for Microbiology, from Sinclair, J.F., et al., 1975 [29]; permission conveyed through Copyright Clearance Center, Inc.

**Figure 5 viruses-15-01308-f005:**
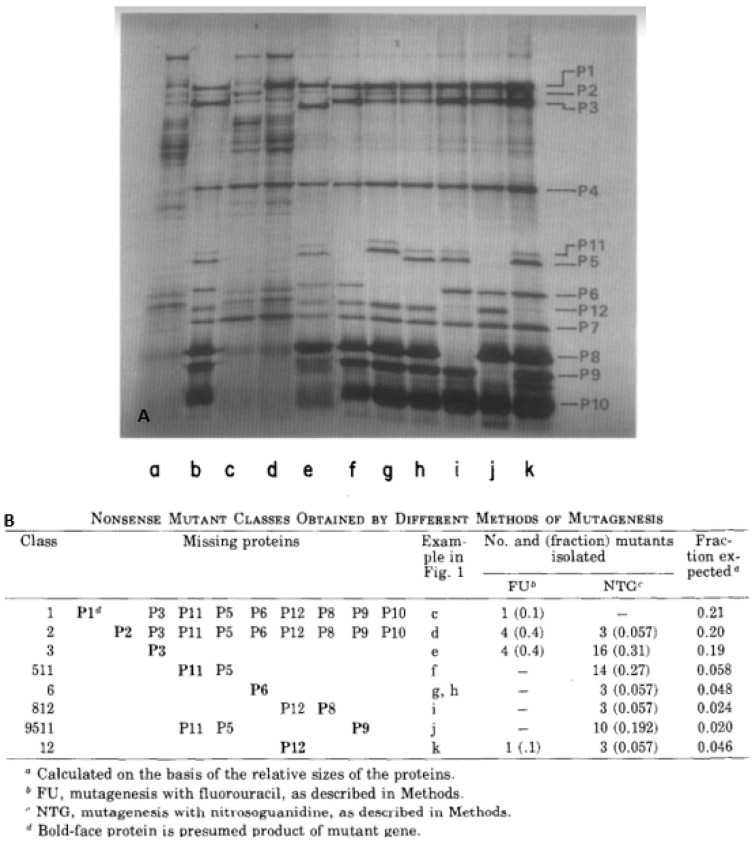
Nonsense mutants as a product of NTG and 5-fluorouracil (FLU) mutagenesis of φ6h1s. The nonsense mutant selection and assignment to the different mutant’s classes based on missing proteins. (**A**) Autoradiogram of the protein synthesis in rifampin treated infected nonpermissive host cells. The samples were radiolabeled with ^14^C—leucine. The protein patterns are labeled a to k. Lane a was the control, noninfected samples, b was the cells infected with φ6h1s. The other patterns were from the bacteriophage mutants. (**B**) Nonsense mutant classes based on the missing proteins. “Figure 1” quote in subfigure B indicates Figure 5, subfigure A. The image and table reprintedd from Ref. [34] Sinclair J.F. et al., 1976, copyright BMC.

**Figure 6 viruses-15-01308-f006:**
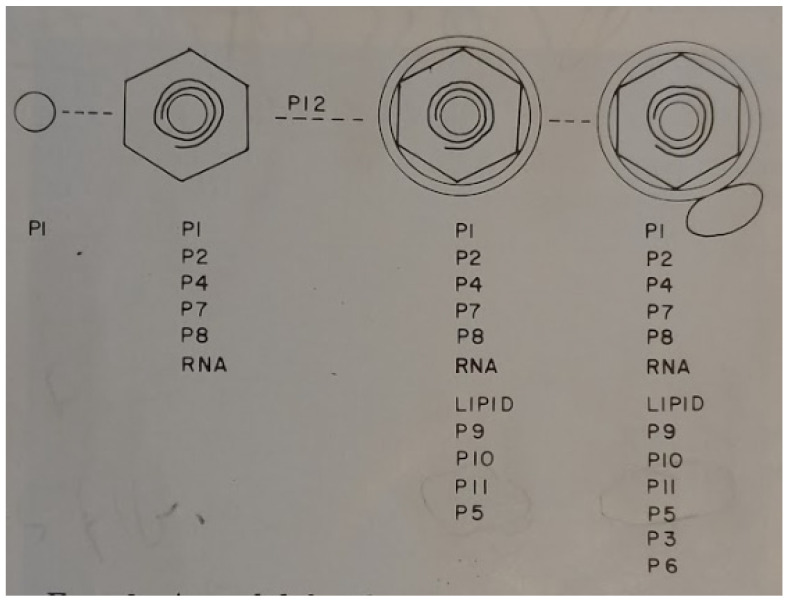
An initial model for structure and assembly of φ6, starting from a core consisting only of P1. Reprinted from Ref. [32] Mindich et al., 1973, copyright BMC.

**Figure 7 viruses-15-01308-f007:**
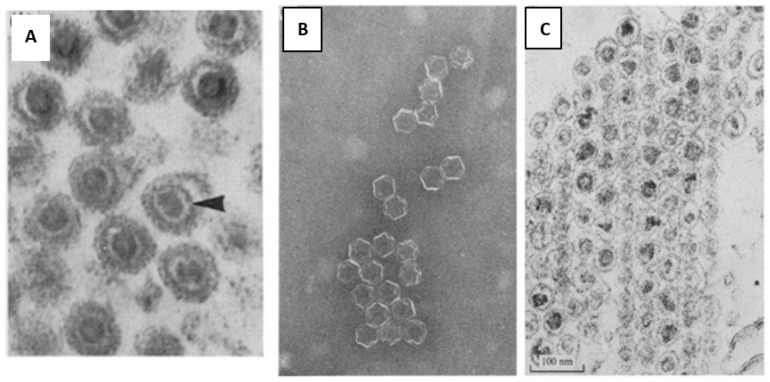
First attempts of structural analysis of phi6. (**A**) The bacteriophage thin section, where the different parts of the head of the phage are clearly seen. The outer diameter of the virion is 65 to 75 nm, inner dense core 30 nm in diameter, then there is another electron dense shell ~50 nm in diameter, which appeared as a dark circle on the micrograph and a bi-lipid membrane of ~7.5 nm thickness. The arrow indicates the 50 nm particle, surrounding the 30 nm core. The outermost dark circle is the phage membrane; (**B**,**C**) Triton X-100 treated phages, (a) A negatively stained preparation where the 45 to 50 nm large rather complex capsid structure is seen. (b) Sectioned pellet of the phage, where the 30 nm core is seen inside the 50 nm particle. Used with permission of Microbiology Society from Bamford et al, 1976 [43], permission conveyed through Copyright Clearance Center, Inc.

**Figure 8 viruses-15-01308-f008:**
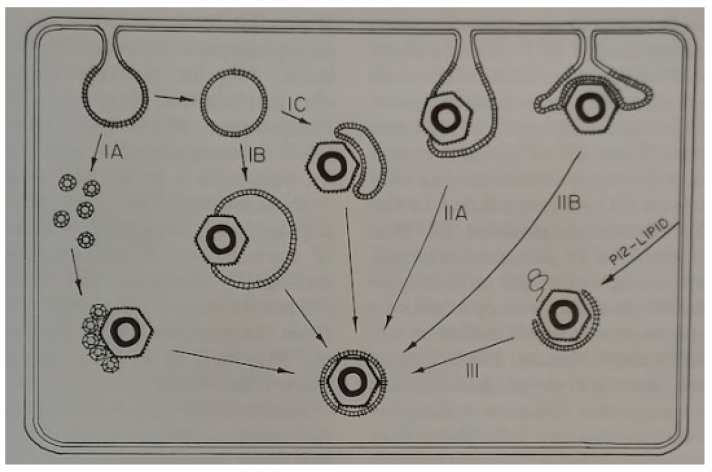
Models of membrane acquisition by bacteriophage φ6. In the Model I, the lipid membrane surrounds the NC within the cell cytosol. In Model II, lipid is derived directly off the inner cell membrane. Reprinted from Ref. [56]. Stitt and Mindich, 1983, copyright BMC.

**Figure 9 viruses-15-01308-f009:**
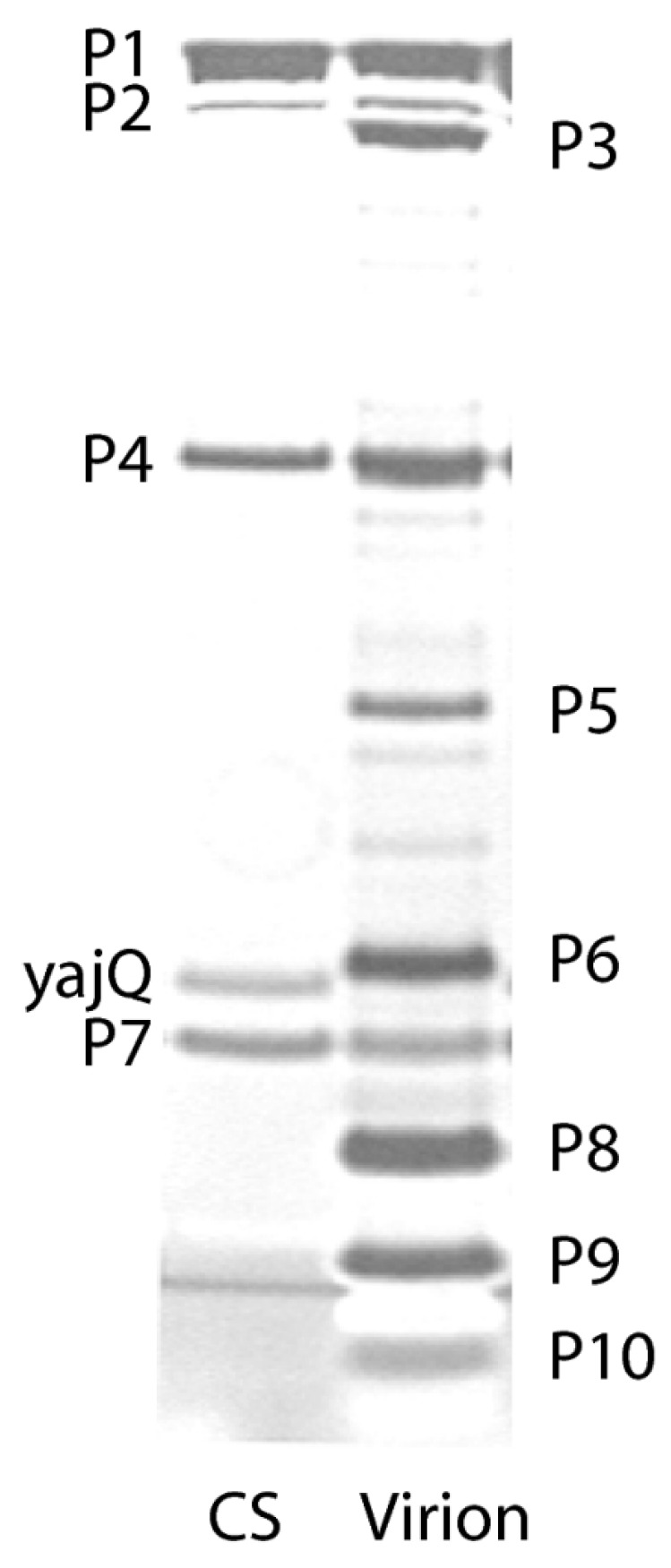
Polyacrylamide gel showing the proteins of the φ6 bacteriophage PC isolated from CS host cells including the YajQ protein. Reprinted from Ref. [61]. Qiao, et.al. 2008, copyright 2008 National Academy of Science.

**Figure 10 viruses-15-01308-f010:**
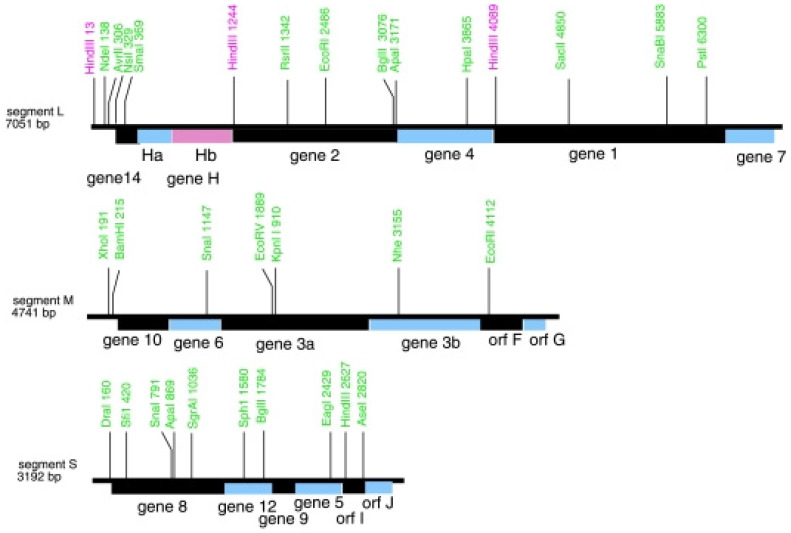
The gene order on the φ8 segments showing the position of gene 7 at the 3′ end and genes Ha and Hb. Orfs I and J are found as shown on the S segment. Reprinted from Ref. [63]. Qiao, et.al. 2009, copyright BMC.

## Data Availability

No new data was created in the writing of the paper.

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
