# Peer review of "Discovery and Classification of the φ6 Bacteriophage: An Historical Review"

_viruses, 2023, doi:10.3390/v15061308_

Round 1

Reviewer 1 Report

The manuscript submitted for review entitled "Discovery and Classification of the φ6 Bacteriophage: an historical review" is a very good historical overview of the discovery and studies related to the φ6 bacteriophage. The manuscript has historical and educational value and is well suited for another type of journal, book chapter, or monograph. The manuscript can be useful for young scientists to understand and learn the steps of identification and study of the viruses, in particular bacteriophages. The manuscript has very little similarity in subject matter to another publication of the authors published in MDPI: "RNA Packaging in the Cystovirus Bacteriophages: Dynamic Interactions during Capsid Maturation".

There are few, insignificant technical errors such as: font of the text, missing description of some abbreviations - EM, spaces, etc.

Sincerely yours,

IS

Author Response

we fixed the technical errors, such as font size, missing coma, missing abbreviations, etc.

Reviewer 2 Report

This review is comprehensively organized and summarized.However,this manuscript only review references published long ago,are there any references published in resently years?

Author Response

This review is historical and only covers the first 10 years of the studies after phi6 discovery. The recent publications are not relevant to the discussion.

Reviewer 3 Report

The authors have done a great job in taking the reader through the historical path of discovery and initial characterization of the first identified cystovirus. The review highlights the discoveries made using crude technology available during that period, which nevertheless provided valuable clues regarding virus composition and assembly. It is also interesting to note that most of these findings were later on confirmed using modern tools and techniques.

Author Response

We have performed the spell check.

Reviewer 4 Report

This is a well-written review that describes the history of bacteriophage φ6 discovery and characterization. My only concern is the limited scope of this historical essay and, consequently, the small audience this work will attract.

I strongly encourage the authors to add a conclusive section focused on the 'the last 10 years of bacteriophage φ6 research'. The way the review ends, it feels like research on this wonderful model system ended with Mindich's experiments in the last 70s. So much has been done since then that the abrupt way in which this review ends is somewhat misleading.

The authors should provide a brief and nicely illustrated description of how modern structural methodologies have capitalized on the original research and unveiled the organization of this unique RNA bacteriophage.

Author Response

We have written an historical review covering the first ten years of phi6 research. Adding the section covering how the modern structural methodologies have capitalized on the original research is out of context. It can be the subject of another review.

Reviewer 5 Report

Title

Discovery and Classification of the phi6 Bacteriophage: an historical review.

Summary

The authors present a review of the early years of phi6 bacteriophage isolation and characterization. Just as people once deduced that the Earth revolves around the sun and the moon revolves around the earth, so too did professors Vidaver and Mindich, and their colleagues, deduce that bacteriophage can have lipid bilayers, procapsids, and dsRNA genomes. 

Major comments

This work is a beautifully-written tribute to the early days of both cystovirus research and the classical elegance of vintage biology. It is a fitting semicentennial marker for phi6 research, appropriate for the journal, and a joy to read.

Author Response

Thank you the positive review.

Round 2

Reviewer 1 Report

The manuscript resubmitted for review entitled "Discovery and Classification of the φ6 Bacteriophage: an historical review" is a very good historical overview of the discovery and studies related to the φ6 bacteriophage. The authors have corrected the technical errors, but these errors were not the basis of my final decision.

The manuscript has historical and educational value. The manuscript can be useful for young scientists to understand and learn the steps to identify and study viruses, especially bacteriophages.

In my opinion, historical reviews are not suitable for this type of specialized scientific journals, because they are historically oriented and do not have their own data (figures, results and conclusions). Therefore, this manuscript, in this version, deserves to be published but in a different form (textbook, book chapter or monograph - combined with the authors' other publications in this field) or in a journal with a different focus.

Sincerely yours,

Author Response

We have addressed the editor's comments and enhanced the conclusion section as directed.